# Lysophosphatidic Acid Receptor Antagonists and Cancer: The Current Trends, Clinical Implications, and Trials

**DOI:** 10.3390/cells10071629

**Published:** 2021-06-29

**Authors:** Yu-Hsuan Lin, Yueh-Chien Lin, Chien-Chin Chen

**Affiliations:** 1Institute of Biomedical Sciences, National Sun Yat-Sen University, Kaohsiung 804, Taiwan; lucaslinyh@gmail.com; 2Department of Otolaryngology, Head and Neck Surgery, Kaohsiung Veterans General Hospital, Kaohsiung 813, Taiwan; 3School of Medicine, National Yang Ming Chiao Tung University, Taipei 112, Taiwan; 4School of Medicine, Chung Shan Medical University, Taichung 402, Taiwan; 5Department of Life Science, National Taiwan University, Taipei 106, Taiwan; 6Vascular Biology Program, Department of Surgery, Boston Children’s Hospital, Harvard Medical School, Boston, MA 02115, USA; 7Department of Pathology, Ditmanson Medical Foundation Chia-Yi Christian Hospital, Chiayi 600, Taiwan; 8Department of Cosmetic Science, Chia Nan University of Pharmacy and Science, Tainan 717, Taiwan

**Keywords:** antagonist, cancer, clinical trial, lysophosphatidic acid, lysophosphatidic receptor, therapy

## Abstract

Lysophosphatidic acid (LPA) is a bioactive lipid mediator primarily derived from membrane phospholipids. LPA initiates cellular effects upon binding to a family of G protein-coupled receptors, termed LPA receptors (LPAR1 to LPAR6). LPA signaling drives cell migration and proliferation, cytokine production, thrombosis, fibrosis, angiogenesis, and lymphangiogenesis. Since the expression and function of LPA receptors are critical for cellular effects, selective antagonists may represent a potential treatment for a broad range of illnesses, such as cardiovascular diseases, idiopathic pulmonary fibrosis, voiding dysfunctions, and various types of cancers. More new LPA receptor antagonists have shown their therapeutic potentials, although most are still in the preclinical trial stage. This review provided integrative information and summarized preclinical findings and recent clinical trials of different LPA receptor antagonists in cancer progression and resistance. Targeting LPA receptors can have potential applications in clinical patients with various diseases, including cancer.

## 1. Introduction

Lysophosphatidic acid (LPA) and LPA receptors (LPARs), including LPAR1 to LPAR6, are integral parts of signaling pathways involved in cellular proliferation/migration/survival, vascular homeostasis, stromal remodeling, lymphocytes trafficking, and immune regulation [1,2,3]. In addition, autotaxin (ATX) is a secreted glycoprotein and functions as a pivotal enzyme to produce extracellular LPA [4,5]. Figure 1 illustrates the extracellular and intracellular biosynthesis of LPA. Consequently, aberrant ATX-LPA-LPAR axis may be involved in the development and progression of many pathologic conditions such as cancer and metastasis [6,7], radio- and chemo-resistances [8,9,10,11,12], fibrotic diseases [13], neuropathic pain [14], arthritis [15], metabolic syndromes [16], and atherosclerosis [17]. Understanding ATX/LPA expression and LPAR-mediated signals elucidated our understanding of the disease mechanisms and highlighted the therapeutic potential of the druggable ATX-LPAR axis. To date, enormous in vivo and in vitro investigations have demonstrated pharmacological antagonization of LPAR to be of paramount significance in reversing pathologic responses. This article sought to update current progress regarding LPAR antagonists in clinical and preclinical settings, emphasizing compounds being evaluated in completed and ongoing clinical trials.

## 2. LPA Receptor-Mediated Signaling in Cancer Biology

ATX-LPA-LPAR signaling is a complex network and intertwines with multiple cellular signaling to contribute a plethora of activities such as proliferation, survival, migration, metastasis, angiogenesis, and inflammation in cancers [6]. Individual LPARs favor different Gα proteins for their downstream signals and cellular functions. In brief, the endothelial cell differentiation gene (EGD) family LPARs (LPAR1 to LPAR3) bind to G_i/o_ and trigger the *Ras**/**Raf**/**MAPK* signaling pathway, phospholipase C (PLC), and the PI3K-Akt pathway [1,3,18,19]. G_q/11_ protein couples LPAR1–5 to mediate PLC and calcium mobilization [20], whereas G_12/13_ interacts with all LPARs, leading to cell migration and invasion through Rho and *Rho*-associated protein kinase (*ROCK*) activation [21]. Signaling through Gs would activate the cAMP-dependent protein kinase A (PKA) signaling pathway and the large tumor suppressor 1 and 2 (*LATS1/2*). It would subsequently inhibit downstream transcriptional co-activators Yes-associated protein (YAP) and PDZ-binding motif (TAZ), which usually drive cancer cell survival, proliferation, invasive migration, and metastasis [22,23]. Interestingly, the *ROCK* activation would suppress LATS1/2 and subsequently activate YAP and TAZ, resulting in tumorigenesis (Figure 2).

The LPA-LPAR signaling pathway is one of the most investigated mechanisms because overexpression of one or more of these receptors was found in several types of cancers. Therefore, the concept to modulate cancer by agonizing or antagonizing LPARs is naturally generated. The following sessions would discuss all LPARs in detail.

### 2.1. LPAR 1

Studies show that LPAR1 enhances metastasis and tumor motility [18]. Aberrant LPAR1 expressions were observed in many cancer cell lines and primary tumors, including ovarian cancer [24], breast cancer [25], liver cancer [26], gastric cancer [27], pancreatic cancer [28,29], lung cancer [30,31], glioblastoma (GBM) [32,33,34], and osteosarcoma [35]. Ovarian cancer is the most investigated cancer in studying the malignancy of LPA signaling. High LPAR1 expressions in ovarian serous cystadenocarcinoma correlate with high proliferation, invasion, migration, and poorer prognosis than those with low expressions [36]. LPAR1 also promotes the development of intratumoral heterogeneity by regulating PI3K/AKT signaling [36]. Retaining the stemness phenotype of ovarian cancer, an autocrine loop via the ATX-LPA-LPAR1-AKT1 signaling axis is critical [37]. In breast cancer, overexpression of LPAR1 in MCF-10A mammary epithelial cells causes cells to acquire an invasive phenotype [38], which correlates with the heparin-binding EGF-like growth factor [39], and mediate basal breast metastasis through LPAR1-PI3K-ZEB1-miR-21 pathways [25]. For hepatocellular carcinoma, LPA-LPAR1 enhances cancer invasion via inducing MMP-9 expression through coordinate activation of PI3K and p38 MAPK signaling cascade [26]. Similarly, increased cancer cell invasiveness mediated by LPAR1 was found in pancreatic cancer [28,29]. For lung A549 cancer cells, the LPAR1/Gi/MAP kinase/NF-κB pathway is involved in LPA-induced oncogenesis, and using the LPAR1/3 antagonist Ki16425 to block LPAR1-mediated signaling would significantly reduce tumor volume [31]. In GBM, LPAR1 expression is also significantly higher than other gliomas [32]. Of interest, the LPA pathway of microglia-and-GBM interaction is a target to improve survival because microglia-derived LPA and ATX upon hypoxia stress may promote GBM proliferation and migration [32]. A recent report indicates LPAR1/PKCα/progesterone receptor pathway is involved in GBM migration [40]. In prostate PC-3 cancer cells, hyperglycemia triggers enhanced vascular endothelial growth factor-C (VEGF-C) expression via the LPAR1/3-Akt-ROS-LEDGF signaling [40]. The LPA-mediated VEGF-C expression can be modified by calreticulin, a multifunctional chaperon protein. In addition, pharmacological LPAR1 receptor antagonism may significantly reduce tumoral lymphatic vessel density and nodal metastasis in tumor-bearing nude mice, suggesting the key role of LPAR1 in prostate cancer lymphatic metastasis [41].

### 2.2. LPAR2

LPAR2 activation has been shown to associate with cell survival because of its anti-apoptosis function. For ovarian cancer, tumors with overexpression of LPAR2 were associated with poorer survivals compared with controls [42]. Furthermore, LPAR2 signaling promotes invasion and metastasis through the production of VEGF [43], EGFR [44], interleukin-8 [45], and urokinase plasminogen activation [46], implying the multiple hyper-vascularization processes. LPAR2-Gi-Src-EGFR-ERK signaling cascade may mediate cell movement and LPA-stimulated COX-2 expression [47]. Together with LPAR1, LPAR2 regulates phosphorylation of ezrin/radixin/moesin (ERM) proteins, known as membrane-cytoskeleton linkers, and leads to promotion of ovarian OVCAR-3 cancer cell migration through cytoskeletal reorganization and formation of membrane protrusions [48]. The metastatic activity of gastric SGC-7901 cells was enhanced as well through LPA-LPAR2-Notch pathway activation [27]. LPAR2 is the major LPAR in colon cancer, and most of the cellular signals by LPAR2 were primarily mediated through interaction with scaffold proteins Na^+^/H^+^ exchanger regulatory factor 2 (NHERF2) [49]. In another two reports, LPA-LPAR2 may facilitate colon cancer proliferation via transcription factor Kruppel-like factor 5 (KLF5) and hypoxia-inducible factor 1α (HIF-1α) activations. The LPAR2 associated HIF-1α expression also promoted breast cancer proliferation/migration and conferred poor prognosis in the Chinese population [50]. Regarding the link between chronic inflammation and cancer, Lin et al. found genetic LPAR2 depletion may attenuate colon cancer development in a colitis mice model triggered by azoxymethane and dextran sulfate sodium [51]. Noteworthy, LPAR2 activation may exert anti-migration effects by blocking EGF-induced migration and invasion of pancreatic Panc-1 cancer cells through the G_12/13_/Rho signaling pathway [52]. G_i2_ protein is also involved in enhanced ovarian cancer invasion and migration via the HIF1α-LPA-LPAR2 axis [24]. The distinct structure of LPAR2 from other LPARs is its carboxyl-terminal tail contains a zinc finger-binding motif to interact with TRIP6 and pro-apoptotic Siva-1. TRIP6 has a PSD95/Dlg/ZO-1 (PDZ)-binding motif to interact with scaffold proteins, particularly NHERF2 [53]. Siva-1 is an early response gene activated by DNA damage that promotes apoptosis through binding up the antiapoptotic Bxl-XL protein. Moreover, Siva-1 acts with p53 and the ubiquitin ligase Mdm2 in the nucleus complexes, and the polyubiquitinated complex would be degraded once the LPA-LPAR2 axis is activated. The functional significance of the LPAR2-activated assembly leads to up-regulation of ERK1/2, PI3K-Akt, and NFκB prosurvival pathways and the subsequent inhibition of apoptosis [54]. LPAR2 can protect cancer cells against apoptotic stress after irradiation and chemotherapy by augmenting DNA damage repair response and inhibiting the mitochondrial apoptosis cascade [55].

### 2.3. LPAR3

LPAR3 is the predominant receptor subtype in colon, liver, and lung cancers. LPAR3-expressing cells significantly promote motility and invasiveness through Ras-, Rac-, Rho-, and PI3K-signaling pathways [20]. In hepatocellular carcinoma, Zuckerman et al. reported distinct LPAR3 expressions within the tumor and normal tissues, and LPAR3 may enhance liver cancer migration via the LPAR3-Gi-ERK/MAPK pathway [56]. Okabe et al. found LPAR3 contributes to hepatocellular carcinoma proliferation and invasion via the β-catenin pathway in rat hepatic RH7777 cancer cells. They also demonstrated that tumor cells with high LPAR3 expression were resistant to cisplatin and doxorubicin through multidrug-resistance-related up-regulation of genes [20]. In melanoma, LPAR3 is essential to promote viability and proliferation, and the Src homology 3 domain is required for LPAR3 to mediate viability in melanoma SK-MEL-2 cells [57,58]. In ovarian cancer, LPAR3 promotes cell expansion and invasion in SKOV-3 cells, and tumors with overexpression of LPAR3 were associated with poor survival [42]. Besides G_q_ and G_i_ proteins, LPAR3 can also activate G_12/13_, increase dephosphorylation and nuclear translocation of YAP, and induce migration of ovarian cancer cells [59]. In addition, the LPA/LPAR3 signaling may initiate mutation-independent epithelial-to-mesenchymal transition (EMT) through β1-integrin-dependent activation of Wnt/β-catenin signaling [60]. Pharmacological suppression of LPAR3 would suppress motility and invasion in various cancers, including hamster pancreatic cancer cells [61], human triple-negative breast cancers [62], fibrosarcoma HT1080 cells, and osteosarcoma HOS cells [63]. Direct targeting of LPAR3 by miR-15b has been shown to repress cell proliferation and drive the senescence and apoptosis of ovarian cancer cells through the PI3K/Akt pathway [64], suggesting the potential mRNA treatment against LPAR3.

### 2.4. LPAR4

In contrast to LPAR1–3, LPAR4 and LPAR5 negatively affected cancer cell proliferation and motility [65]. LPAR4 attenuates tumor motility and colony formation in colon cancer cell lines. Knockdown of LPAR4 in the long-term 5FU treated DLD1 cells increased cell motility [66,67]. Similarly, LPAR4 depletion increases tumor motility in pancreatic cancer cells [65] and increases tumor proliferation in head and neck carcinoma [68]. Another recent study by Eino et al. found that LPAR4 is critical for developing a fine capillary network in brain tumors [69]. LPAR4 promotes endothelial cell-cell adhesion and VCAM-1 expression via RhoA/ROCK signaling, which may enhance anti-PD1 therapy efficacy and lymphocyte infiltration [69]. However, a contradictory pro-tumorigenesis was found in fibrosarcoma. In HT1080 cells, LPAR4 promotes cell invasion and invadopodium formation via cAMP/EPAC/Rac1 signaling [70]. Of interest, LPAR4/6 are necessary for embryogenic angiogenesis to activate YAP and transcriptional coactivator TAZ via the G_12_/G_13_ signaling pathway [71]. In the malignancies, YAP promotes cancer proliferation and migration in bladder cancers through YAP-Mask2 [72] and lung cancers through LKB1-YAP-human telomerase RAN (hTERC), respectively [73]. These suggested the involvement of LPAR4 in YAP-mediated cancer progression.

### 2.5. LPAR5

LPAR5 was considered a negative regulator in cancer cell motility and survival [69]. The inhibitory effect of LPAR5 on cell motility has been shown in pancreatic cancer [69] and sarcoma [74]. Nevertheless, contradictory effects of LPAR5 were found in different cancers. Okabe et al. reported upregulation of the *LPAR5* gene with aberrant unmethylated status enhanced cell proliferation and motility in rat liver-derived hepatoma RH7777 and lung-derived adenocarcinoma RLCNR cells [75]. Blocking LPAR5 in thyroid cancer with a selective LPA5 antagonist TCLPA5 attenuated cancer proliferation and migration via PI3K/Akt signaling in vivo and in vitro [76]. Moreover, depletion of LPAR5 in murine B16-F10 melanoma resulted in fewer lung metastasis [77]. Interestingly, LPAR5 appears to mediate chemorepulsion in response to LPA. The underlying mechanism was proposed to be mediated via a non-canonical elevation of cAMP along with reduced PIP3 signaling in melanoma B16 cells [78]. LPAR5 expression is markedly increased in long-term cisplatin-treated melanoma cells [8]. Therefore, LPAR5 knockdown significantly conferred chemo-resistance and enhanced cancer cell survival [8]. In addition to the cancer cell growth and metastasis, LPAR5 was shown to suppress the function of CD8-positive cytotoxic T cells by inhibiting intracellular Ca^2+^ mobilization and ERK activation, suggesting LPAR5 might act as a mediator of immune suppression [79].

### 2.6. LPAR6

Reports regarding LPAR6 in cancer are relatively limited compared with other LPARs [2]. Several articles investigated the role of LPAR6 in liver, pancreatic, and colon cancers. LPAR6 expression in hepatocellular carcinoma correlated with poorer survival [80] and increased microvascular invasion [81]. Moreover, LPAR6 promotes hepatocellular carcinoma proliferation via the NCOA3-LPAR6-HGF signaling cascade, and the tumor-suppressive effect by depletion of LPAR6 is similar to that of anti-HGF treatment [82]. In pancreatic cancer, LPAR6 knockdown also inhibited cancer invasion and colony formation [67]. However, LPAR6 can, by contrast, be a negative regulator in different cancers. LPAR6 knockdown caused the formation of larger colonies [83] and enhanced motility in colon DLD1 and HCT116 cancer cells [67]. The role of LPAR6 in various cancer types should be further characterized in the future.

## 3. LPARs and Cancer Resistance to Chemotherapy and Radiation

Radiation therapy and chemotherapy are both primary cancer treatments. However, cancers often developed resistance and long-term side effects after radiotherapy and chemotherapy. Therefore, it is crucial to understand the molecular and physiological changes in patients with cancer receiving these therapies, which will help develop a better treatment for the patients. Remarkedly, the *ENPP2* gene, which encoded LPA generating enzyme ATX, was found to serve as one of 90 drug-resistance genes [84]. Up-regulation of the ATX-LPA axis was found in refractory diseases treated by radiation and chemotherapy [85]. These results suggested ATX-LPA axis might be involved in chemo- and radio-resistances in cancers.

Chemotherapies such as docetaxel and doxorubicin would induce cancer cell apoptosis through ceramide formation [86,87]. Ceramide, belonging to sphingolipids and converted to sphingosine by ceramidases, promotes apoptosis by releasing cytochrome C from mitochondria and activates caspases. The pro-apoptotic effects of ceramide are counteracted by LPA-LPAR1-induced nuclear factor erythroid-2-related factor-2 (Nrf2) stabilization, which upregulates multidrug-resistant transporters and antioxidant proteins [88]. Another pathway of LPA-mediated chemo-resistance is to enhance sphingosine 1-phosphate (S1P) by activating phospholipase D (PLD) [89] and increasing sphingosine kinase 1 (Sphk1) [90]. Histone deacetylase (HDAC) activation is also involved in chemo-resistance. LPA activated HDAC and subsequently prevented cancer apoptosis which was induced by HDAC inhibitors [91]. Therefore, targeting ATX was considered the center to overcome drug resistance. For example, ONO-8430506, an ATX inhibitor, had a synergic effect with doxorubicin to reduce tumor growth and lung metastasis in an orthotopic mice breast cancer model [92]. In addition, novel benzene-sulfonamide analogues acting as ATX antagonists can reduce paclitaxel resistance in 4T1 murine breast cancer cells and B16 murine melanoma cells [93].

Besides ATX antagonists, targeting individual LPARs also has potency against resistance [94]. Recent studies emphasized the effects of LPAR2 and LPAR5 on drug resistance. Interestingly, LPAR2 promoted the acquisition of chemo-resistance, whereas LPAR5 suppressed chemo-protection [9,10]. Administration of the LPAR2 agonist, GRI-977143, in human lung A549 adenocarcinoma cells with long-term cisplatin treatment increased cell survival rate [10]. Similar results for LPAR2 mediated cisplatin-resistance was found in melanoma A375 cells [11] and fibrosarcoma HT1080 cells [12]. The Gi and LPAR2 interacting proteins, TRIP6 and NHERF2, were believed to involve LPAR2-mediated chemo-resistance [12,54]. Furthermore, LPAR2-mediated small GTPase Arf6 activation contributes to Sunitinib and Temsirolimus resistance in renal 786-O renal cell carcinoma cells [95]. Silencing of ArfGAPs, AMAP1, and EPB41L5 enhanced the drug sensitivity, suggesting Arf6-mediated LPAR2 recycling might be the key for chemotherapy resistance. Interestingly, the decline of G protein signaling regulators (RGS) might also participate in LPA-mediated chemo-resistance via reducing de-activation of *G protein-coupled receptors* (GPCR). For example, RGS10 and RGS17 knockdown conferred cisplatin-resistance in ovarian cancer cells [96], whereas increasing RGS17 in nasopharyngeal cancer improves sensitivity to 5-fluorouracil [97]. The detailed mechanism of LPAR5 is not clarified, so further investigation is required. Notably, LPAR2 is imperative to protect normal organs, especially intestinal epithelium and myeloid progenitors, against ceramide-induced cell death. Deng et al. found the LPA administration protected IEC-6 cells from camptothecin-induced apoptosis through inhibiting caspase-3 activation mediated by the attenuation of caspase-9 activation [55]. The role of LPAR3 in chemo-resistance remains controversial. A selective LPAR3 agonist, OMPT, would reduce the cell viability of cisplatin-treated lung A549 adenocarcinoma cells [10]. Another article indicated that LPAR3 conferred chemo-resistance by upregulating multidrug resistance-related genes because the cell survival in LPAR3-expressing rat hepatoma cells treated with cisplatin or doxorubicin was higher than controls [20]. Conclusively, the above information provides a horizon to ameliorate drug resistance by targeting LPAR2, LPAR3, and LPAR5 in different cancers.

For radiotherapy, it is therapeutically important to lower the radio-resistance of cancer cells but to protect normal tissue injury from radiation. In response to radiation injury, adipose tissue increased ATX production and LPARs expressions to enhance LPA signaling [98]. Since ATX responses were found to precede the radiation-induced inflammatory cascade in several in vivo studies [99], the ATX-LPA-inflammatory cycle may play a vital role in desensitizing cancer cells to radiotherapy. Different LPARs mediate radio-resistance in different cancer cells, and LPAR1 is a pronounced receptor to mediate radio-resistance because of its ability to stabilize Nrf2 via PI3K signaling [86,88,98]. Nrf2 mediates multiple antioxidant enzymes, such as glutathione S-transferase A2 (GSTA2) and NADPH quinone oxidoreductase 1 (NQO1), to protect cancer cells from oxidative stress [100]. Nrf2 over-expression enables cancer cells against radiation injury by cross-talk with multiple LPA-induced DNA repair proteins expressions (e.g., ATM, ATR, PARP-1) to activate NF-κB signaling [101,102]. It implies that Nrf2 destabilization may be a potential strategy to overcome radio-resistance [103].

Interestingly, not only ATX, but LPAR2 expression are also upregulated in response to radiation. Radiation-induced DNA double-strand breaks via ATM-mediated NF-κB activation, resulting in increment of serum ATX/LPA levels and subsequently LPAR2 activation to accelerate γH2AX histones resolution [101]. In breast cancer adipose tissue, Meng et al. found radiation-induced LPAR1, LPAR2, ATX, cyclooxygenase-2, and numerous inflammatory mediators increased in response to γ-radiation ranging from 0.25 to 5 Gy, resulting in blunting radio-sensitivity of breast cancer cells [98]. LPAR2 is imperative to protect normal organs, especially intestinal epithelium and myeloid progenitors, against ceramide-induced cell death. In a radiation-exposed murine model, intraperitoneal administration of synthetic LPA, octadecenyl thiophosphate (OTP), may reduce radiation-associated mortality under lethal dose (6–12 Gy) [104]. By contrast, LPAR2 knockout mice (LPAR2^−/−^) can no longer be rescued from radiation-induced apoptosis upon OTP administration. In another whole-body radiation mice model, administration of the LPAR2 agonist, GRI977143, significantly increased mean survival by saving apoptotically condemned cells from radiation-induced cell death [105]. In contrast, DBIBB, a more specific LPAR2 agonist without binding affinity to other LPARs, was found to alleviate the acute hematopoietic and gastrointestinal radiation responses for wild-type C57BL/6 mice when drug treatment was delayed to 72 h post-irradiation. In human hematopoietic progenitors, DBIBB significantly enhanced cell survival and the differentiation of the myeloid cell lineage after irradiation [106]. Moreover, DBIBB treatment increased the post-radiation survival of the rat intestinal crypt epithelium-like cell line IEC-6 cells by augmenting DNA repair via LPAR2 signaling [106].

## 4. LPA and Chemotherapy-Induced Neuropathic Pain (NP)

Cancer patients treated with chemotherapy, such as paclitaxel, usually suffer with severe peripheral neuropathic pain [14]. Mechanisms that underlie NP are complex for multiple pathways activated through inflammatory molecules, growth factors, lipid metabolites, and cellular responses in peripheral and central nervous systems. Aberrant LPA production and signaling were pivotal for NP initiation in which the LPA-LPAR1 axis mediated peripheral mechanisms [107], including dorsal root demyelination, PKCγ, calcium channel subunit α2δ1 up-regulation [107], neuronal growth cones retraction, and morphological changes of Schwann cells [108]. LPAR3, together with LPAR1, was also responsible for NP by amplifying central LPA production via glial cells [109]. Interestingly, Uchida et al. found that LPA production significantly increased within 24 h after the first paclitaxel treatment [14]. The paclitaxel-induced mechanical allodynia disappeared in LPAR1- and LPAR3- knockout mice [14], as well as mice pretreated with LPAR1/3 antagonist *Ki16425* [110]. Moreover, significant attenuation of nerve demyelination was also observed in LPAR1-/LPAR3- knockout mice [110]. In addition, the LPAR5 signaling was shown to enhance microglial migration/cytotoxicity and induced a distinct pro-inflammatory signature via the protein kinase D (PKD) pathway [111]. In summary, antagonizing LPAR1/3/5 may be a promising strategy to manage chemotherapy-induced NP.

## 5. Application of LPAR Agonist/Antagonist in Cancer

Many LPAR antagonists were developed to attenuate proliferation, progression, invasion, and metastasis in some preclinical studies. BrP-LPA, a pan-LPAR antagonist, was used to treat breast MDA-MB-231 cancer cells, and Zhang et al. found more decreased intra-tumoral blood vessel density treated by BrP-LPA (10 mg/kg) compared with those by paclitaxel (10 mg/kg) [112]. Through LPAR2, BrP-LPA may also sensitize vascular endothelial cells in mice GL-261 glioma cells to improve malignant glioma response to radiation therapy [113]. Furthermore, Ki16425 is one of the most investigated agents targeting both LPAR1 and LPAR3, and its anti-tumor effects were validated in several cancer cells [31,114,115]. As discussed above, Ki16425 may benefit the reduction of chemotherapy-induced NP [110]. A more potent dual LPAR1/3 antagonist, Debio-0719, was found to reduce pulmonary and bone metastases of murine 4T1 breast cancer cells without affecting primary tumor size [116]. Similar results were observed in the MDA-MB-231T experimental metastasis mouse model, suggesting Debio-0719, a potent therapeutic agent to prevent breast cancer metastasis and induce dormancy at secondary tumor sites [117]. Another LPAR1/3 antagonist, Ki16198, effectively suppressed pancreatic cancer invasion and metastasis partially through inhibiting MMP production [118].

Other LPAR antagonists were less discussed, while H2L5186303 for LPAR2 attenuated fibrosarcoma HT1080 cells invasiveness [119]. The 4-methylene-2-octyl-5-oxotetra-hydrofuran-3-carboxylic acid (C75) and xanthenylacetic acid (XAA) for LPAR6 ameliorated hepatocellular carcinoma growth through modulating mitochondria homeostasis and arresting cancer cells at the G1-phase cell cycle [120].

As mentioned above, targeting LPAR5 was proposed to be a good option against cancer development in certain cancers [8,9,65]. LPAR5 antagonist TCLPA5 attenuated cancer proliferation and migration in thyroid cancer [76]. Moreover, depletion of LPAR5 in murine B16-F10 melanoma resulted in fewer lung metastasis [77], suggesting pharmaceutic inhibition of LPAR5 may also manage melanoma-mediated metastasis. On the other hand, since LPAR5 contributes to NP by affecting microglia biology and induces a distinct pro-inflammatory phenotype [111], various antagonists such as compound 7e [121] and AS2717638 [122] were considered to control chemotherapy-induced NP [76,111].

## 6. Clinical Trials of LPAR Antagonists

Since the extensive involvement of LPARs in cancers, questions have emerged regarding their clinical significance. However, as of now, there are no therapeutic clinical trials of LPAR antagonists in cancer therapy. A diagnostic trial (NCT00986206) enrolled 525 patients with ovarian cancer, or at risk for ovarian cancer, and aimed to develop a serum- or plasma-based assay to quantitate LPA levels in the early detection of ovarian cancer [123], but there was no conclusive result.

Apart from cancer, LPAR1 antagonists, BMS-986020, and BMS-986278 focused on idiopathic pulmonary fibrosis (IPF) and aimed to prove the application of LPAR. In the case of BMS-986020, clinical trials including NCT01766817 (Phase 2) [124,125], NCT02068053 (Phase 1) [126], and NCT02101125 (Phase 1) [127] were completed. The NCT01766817 study revealed that BMS-986020 treatment significantly slowed the pulmonary function decline compared with placebo [124]. Regarding BMS-986278, the Phase 1 study, NCT03429933, was completed [128], and a Phase 2 trial, NCT04308681, is recruiting participants [129]. Interestingly, there is a diagnostic trial recruiting IPF patients to validate the safety, tolerability, kinetics, and repeatability of an LPAR1 PET ligand 18F-BMS-986327 [130]. Moreover, a Phase 2 study (NCT01651143) focused on SAR-100842, a potent, selective oral antagonist of the LPAR1, for diffuse cutaneous systemic sclerosis. The results suggested that SAR-100842 is safe, moderately effective, and well-tolerated in patients, whereas a further larger controlled trial is required to confirm the clinical efficacy [131]. Table 1 summarizes the clinical trials targeting LPARs. 

## 7. Limitation of LPAR Antagonist in Cancer

Despite the apparent relevance of LPA signaling in cancer initiation, progression, metastasis, and developments of resistance against chemo- and radio-induced cancer cell death, no inhibitors targeting LPARs have progressed to cancer-related clinical trials thus far. One possible reason is that ATX, lipid phosphate phosphatase, and non-GPCR LPARs signaling have been implicated in tumor growth and metastasis, suggesting the limitation by attenuating LPA-GPCR signaling only. Another reason is that the LPA receptor subtypes might exert distinct effects depending on the type and cellular origin of the individual carcinoma and thereby complicate the use of LPAR antagonists. The third reason is that LPAR antagonists might have cross-activities on multiple receptors and other key targets, making individuals more complicated by jeopardizing physiological activities that required specific signaling. Nevertheless, clinical trials of LPAR antagonists on cancer research could be anticipated to be conducted in the future, especially concerning its role as adjuvants to traditional chemo- and/or radio-therapy to reduce resistance, prolong cancer remission, and lower treatment-associated adverse events. Additionally, LPA signaling antagonism could have high clinical significance to minimize side effects, such as drug-induced pain, acute radiation syndrome, or radiation-induced fibrosis. To conclude, further perspectives should shed light on precise targeting of particular LPARs in each cancer to make LPAR antagonists clinically beneficial, thus improving the prognosis of cancer patients.

## 8. Conclusions

In conclusion, it deserves our attention that multiple therapeutic agents undergo clinical trials or preclinical evaluation for various diseases via inhibition of LPA signaling. Their safety is generally acceptable, and the LPAR antagonists are potentially effective and novel for improving pain and current cancer therapies. In general, being inflammatory mediators, LPA signaling inhibitors could be potential therapeutic modalities for chemoprevention, enhancing the efficacy of chemotherapy and radiotherapy and improving prognosis.

## Figures and Tables

**Figure 1 cells-10-01629-f001:**
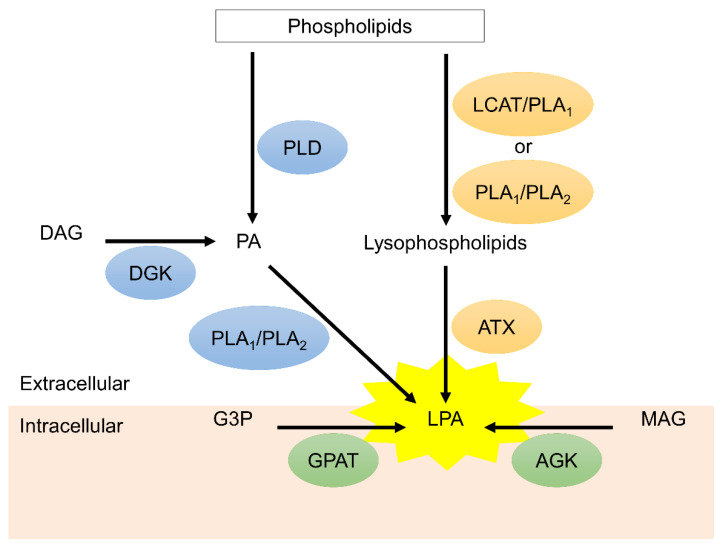
The biosynthesis of lysophosphatidic acid (LPA). Extracellular LPA is mainly synthesized by autotaxin (ATX) via conversion of lysophospholipids, which are hydrolyzed from phospholipids through lecithin-cholesterol acyltransferase (LCAT)/phospholipase A1 (PLA_1_) or PLA_1_/PLA_2_ mechanism. Phosphatidic acid (PA) on the plasma membrane is another resource of extracellular LPA. PA can be generated from phospholipids and diacylglycerol (DAG) via phospholipase (PLD) and diacylglycerol kinase (DGK), respectively. PA is converted to LPA through PLA_1_ and PLA_2_. On the other hand, monoacylglycerol (MAG) and glycerol 3-phosphate (G3P) on the mitochondria generated intracellular LPA via acylglycerol kinase (AGK) and glycerophosphate acyltransferase (GPAT), respectively.

**Figure 2 cells-10-01629-f002:**
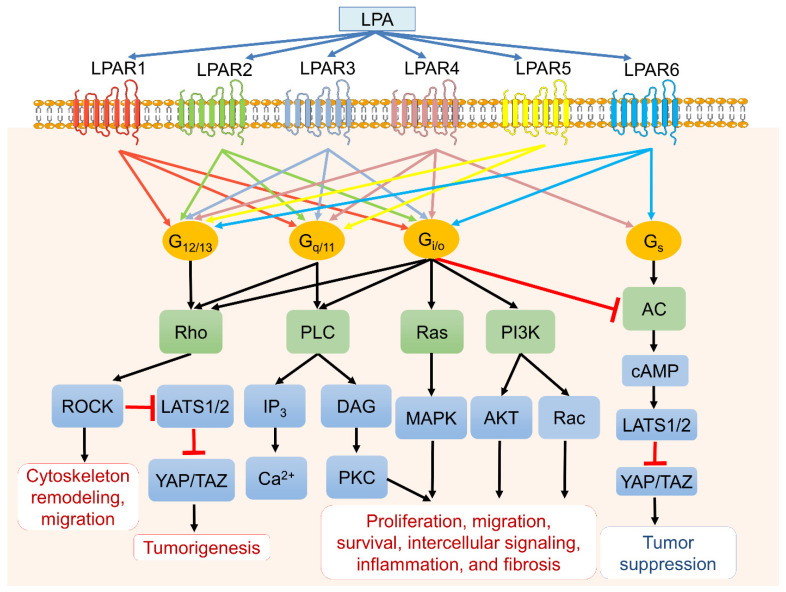
LPA, LPA receptors (LPARs), and downstream signaling pathways. LPA binds six primary LPA transmembrane receptors (LPAR1 to LPAR6) with varying affinities that couple to four different heterotrimeric G proteins (G_12/13_, G_q/11_, G_i/o_, and G_s_) and trigger various downstream signaling cascades. LPA subsequently mediates cellular events such as cell proliferation, survival, apoptosis, migration, cytoskeleton reorganization, fibrosis, and inflammation.

**Table 1 cells-10-01629-t001:** A summary of clinical trials targeting LPA receptors (LPARs).

No.	ClinicalTrials.gov Identifier	Mechanism	Project Title	Study Design	Outcome
1	NCT01766817	LPAR1 antagonist (BMS-986020)	Safety and efficacy of a lysophosphatidic acid receptor antagonist in idiopathic pulmonary fibrosis	Phase 2; parallel-arm, multicenter, randomized, double-blind, placebo-controlled trial; 143 patients with idiopathic pulmonary fibrosis were randomized and treated.	BMS-986020 600 mg bid treatment for 26 weeks significantly slowed the lung function decline compared with placebo [124,125].
2	NCT02068053	LPAR1 antagonist (BMS-986020)	Absorption, distribution, metabolism, and excretion (ADME) study of BMS-986020	Phase 1; a single group assignment to investigate the pharmacokinetic, biotransformation, routes of elimination, and mass balance of BMS-986020 in humans; 6 healthy participants.	It was completed in April 2014. No results were posted [126].
3	NCT02101125	LPAR1 antagonist (BMS-986020)	Drug interaction study with Rosuvastatin	Phase 1; an open-label, single-sequence study to evaluate the effect of concomitant administration of BMS-986020 on the single-dose pharmacokinetics of Rosuvastatin in healthy subjects; 26 healthy participants.	It was completed in May 2014. No results were posted [127].
4	NCT03429933	LPAR1 antagonist (BMS-986278)	A study of experimental medication BMS-986278 given to healthy participants	Phase 1; a double-blind, placebo-controlled, randomized, single and multiple ascending dose study of oral BMS-986278 administration in healthy participants; 112 healthy participants.	It was completed in March 2019. No results were posted [128].
5	NCT04308681	LPAR1 antagonist (BMS-986278)	A study measuring the effectiveness, safety, and tolerability of BMS-986278 in participants with lung fibrosis	Phase 2; a multicenter, randomized, double-blind, placebo-controlled study; 360 patients with lung fibrosis.	It started in July 2020 and is currently still recruiting [129].
6	NCT04069143	LPAR1 tracer (BMT-136088)	Safety, tolerability, kinetics, and repeatability of the novel LPA1 PET ligand 18F-BMS-986327	Phase 1; an open-label study; 20 participants (healthy or with idiopathic pulmonary fibrosis).	It started in October 2019 and is currently still recruiting [130].
7	NCT01651143	LPAR1 antagonist (SAR100842)	Proof of biological activity of SAR100842 in systemic sclerosis	Phase 2; a double-blind, randomized, placebo-controlled study; 32 patients with diffuse cutaneous systemic sclerosis.	SAR100842 was well tolerated in patients. The modified Rodnan skin thickness score improved during the study, although the difference was not significant [131].

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
