# Peer review of "Lysophosphatidic Acid Receptor Antagonists and Cancer: The Current Trends, Clinical Implications, and Trials"

_cells, 2021, doi:10.3390/cells10071629_

Round 1

Reviewer 1 Report

The authors put together a nice manuscript detailing the differential roles that ATX-LPAR signaling axis play in regulating cancer progression. The perspectives and arguments written regarding the use of compounds targeting this signaling axis to potentially treat cancer is also well presented. However, there are several issues detailed below that requires revision.

Line 45: It should be “druggable ATX-LPAR axis”?

Line 92-94: Is this statement correct? I believe the findings from Dr. Xu’s group point to the role of PPARg in regulating ZIP4 expression and not via LPAR1 – since the LPAR1/3 antagonist Ki16425 did not block LPA-induced Z1P4 expression.

Line 121: What is ERM? Please define.

Line 132-133: What do you mean by “LPAR2 activation may exert against tumor”? Do you mean activation of LPAR2 may exert anti-migratory effects by blocking EGF-induced migration?

Line 135: Where does it show in Ref 54 that Gai is involved in HIF1a-LPA-LPAR2 mediated ovarian cancer invasion? Is there a missing reference?

Line 153: Should be LPAR3-Gai-ERK/MAPK pathway.

Line 192 to 197: There appears to be a disconnect in these sentences. These sentences should be compiled in the following manner. “Interestingly, LPAR5 appears to mediate chemorepulsion in response to LPA. The underlying mechanism was proposed to be mediated via a non-canonical elevation of cAMP along with reduced PIP3 signaling in melanoma B16 cells [77]. LPAR5 expression is markedly increased in long-term cisplatin-treated 193 melanoma cells. LPAR5 knockdown significantly conferred chemo-resistance and enhanced cancer 196 cell survival [8].”

Line 301: Typo – LPAR2

Line 333: What do you mean by the functional de-sensitization might be promising? Promising for?

Line 363: Please rephrase this sentence.

Line 419: I do not understand this statement. What do you mean when you say "LPAR antagonist are applicable to treat various cancer because they majorly impact the tumor microenvironment, which is relatively not specific for particular types of cancer cells." What do you mean by relatively not specific? Will these nonspecific effects of LPAR antagonist in the tumor microenvironment limit its use as cancer therapy?

Author Response

Reviewer 1:

The authors put together a nice manuscript detailing the differential roles that ATX-LPAR signaling axis play in regulating cancer progression. The perspectives and arguments written regarding the use of compounds targeting this signaling axis to potentially treat cancer is also well presented. However, there are several issues detailed below that requires revision.

  1. Line 45: It should be “druggable ATX-LPAR axis”?
  • Thank you for your thoughtful recommendation. Yes, it should be ATX-LPAR axis. The revised parts were highlighted in red color in the revised manuscript. We’re grateful for your sincere comments.
  1. Line 92-94: Is this statement correct? I believe the findings from Dr. Xu’s group point to the role of PPARg in regulating ZIP4 expression and not via LPAR1 – since the LPAR1/3 antagonist Ki16425 did not block LPA-induced Z1P4 expression.
  • Thank you for your professional recommendation. We greatly agree with your comments and have deleted our misleading description, since the session focuses on LPAR1. We’re grateful for your scientific comments.
  1. Line 121: What is ERM? Please define.
  • Thank you for your sincere recommendation. We have added the description as follows, “Together with LPAR1, LPAR2 regulates phosphorylation of ezrin/radixin/moesin (ERM) proteins, known as membrane-cytoskeleton linkers, and leads to promote ovarian OVCAR-3 cancer cell migration through cytoskeletal reorganization and formation of membrane protrusions [50].”All revised parts were highlighted in red color in the revised manuscript. We’re grateful for your sincere comments.
  1. Line 132-133: What do you mean by “LPAR2 activation may exert against tumor”? Do you mean activation of LPAR2 may exert anti-migratory effects by blocking EGF-induced migration?
  • Thank you for your professional recommendation. We have re-written the description as follows to make it clear, “Noteworthy, LPAR2 activation may exert anti-migration effects by blocking EGF-induced migration and invasion of pancreatic Panc-1 cancer cells through the G12/13/Rho signaling pathway [55].”
  • All revised parts were highlighted in red color in the revised manuscript. We’re grateful for your sincere comments.
  1. Line 135: Where does it show in Ref 54 that Gai is involved in HIF1a-LPA-LPAR2 mediated ovarian cancer invasion? Is there a missing reference?
  • You’re correct. We have updated the reference with the following, “Lysophosphatidic acid stimulates epithelial to mesenchymal transition marker Slug/Snail2 in ovarian cancer cells via Gαi2, Src, and HIF1α signaling nexus. Oncotarget. 2016 Jun 21;7(25):37664-37679.” The reference is precisely suitable. We’re grateful for your professional comments.
  1. Line 153: Should be LPAR3-Gai-ERK/MAPK pathway.
  • Thank you for the correction. We revised it as LPAR3-Gi-ERK/MAPK pathway. We’re grateful for your sincere comments.
  1. Line 192 to 197: There appears to be a disconnect in these sentences. These sentences should be compiled in the following manner. “Interestingly, LPAR5 appears to mediate chemorepulsion in response to LPA. The underlying mechanism was proposed to be mediated via a non-canonical elevation of cAMP along with reduced PIP3 signaling in melanoma B16 cells [77]. LPAR5 expression is markedly increased in long-term cisplatin-treated 193 melanoma cells. LPAR5 knockdown significantly conferred chemo-resistance and enhanced cancer 196 cell survival [8].”
  • Thank you for your kind and professional comments. We have re-written the session according to your suggestion. All revised parts were highlighted in red color in the revised manuscript. We’re really grateful for your sincere comments.
  1. Line 301: Typo – LPAR2
  • Thank you for your thoughtfulness. We have corrected it as LPAR2.
  1. Line 333: What do you mean by the functional de-sensitization might be promising? Promising for?
  • Thank you for your professional comments. Since neuropathic pain is too general to discuss, we have deleted the description you mentioned and re-written the whole session to focus on the ATX-LPAR axis in chemotherapy-induced neuropathic pain. All revised parts were highlighted in red color in the revised manuscript.
  1. Line 363: Please rephrase this sentence.
  • Thank you for your professional recommendation. We have rewritten the whole paragraph of targeting LPAR5 in cancer therapy. All revised parts were highlighted in red color in the revised manuscript. We’re grateful for your comments.
  1. Line 419: I do not understand this statement. What do you mean when you say "LPAR antagonist are applicable to treat various cancer because they majorly impact the tumor microenvironment, which is relatively not specific for particular types of cancer cells." What do you mean by relatively not specific? Will these nonspecific effects of LPAR antagonist in the tumor microenvironment limit its use as cancer therapy?
  • Thank you for your scientific recommendation. The statement was ambiguous and not conclusive. After discussion, we decided to delete the description to make our conclusion more objective and easier understood.

Reviewer 2 Report

This review represents a nice update about LPAR antagonists and trials in different pathologies, not only cancer.

In figure legend 1 include DGK function.

In figure two: G proteins are trimeric. The subunit alpha and the dimer beta-gamma activate distinct effectors; to avoid mistakes erase the alpha letter from each G protein and leave it as G12/13, Gq/11, Gi/o or Gs.

Add this referenceOncogene volume 37, pages1457–1471 (2018), in page 3 lane 88 when you refer to aberrant LPAR1 expression in glioblastoma, also the next reference: Cells, 2021 Apr 4;10(4):807, doi: 10.3390/cells10040807 about LPA1R/PKCa/PR pathway involved in cell migration.

LPA binding to PPARgamma is a non-classical mechanism that doesn´t involve LPAR1, so I recommend moving or take out the statement in lanes 90-94 pages 3 and 4 ref [37]. 

 Lane 121 page 4, define ERM.

If you want to include neuropathic pain and pulmonary fibrosis in this manuscript, change the review title, including other pathologies.

Author Response

Reviewer 2:

This review represents a nice update about LPAR antagonists and trials in different pathologies, not only cancer.

  1. In figure legend 1 include DGK function.
  • Thank you for your professional recommendation. We have revised and rewritten the figure 1 legend to introduce DGK function. All revised parts were highlighted in red color in the revised manuscript. We’re grateful for your sincere comments.
  1. In figure two: G proteins are trimeric. The subunit alpha and the dimer beta-gamma activate distinct effectors; to avoid mistakes erase the alpha letter from each G protein and leave it as G12/13, Gq/11, Gi/o or Gs.
  • Thank you for your professional recommendation. We have revised figure 2 and the whole G proteins in our manuscript according to your recommendation. All revised parts were highlighted in red color in the revised manuscript.
  1. Add this reference: Oncogenevolume 37, pages1457–1471 (2018), in page 3 lane 88 when you refer to aberrant LPAR1 expression in glioblastoma, also the next reference: Cells, 2021 Apr 4;10(4):807, doi: 10.3390/cells10040807 about LPA1R/PKCa/PR pathway involved in cell migration.
  • Thank you for your recommendation. We have added these two references into our revised manuscript highlighted in red color. We’re grateful for your comments.
  1. LPA binding to PPARgamma is a non-classical mechanism that doesn´t involve LPAR1, so I recommend moving or take out the statement in lanes 90-94 pages 3 and 4 ref [37]. 
  • Thank you for your professional recommendation. We agree with it. So, we removed the reference and deleted the sentence, ”Moreover, LPA promotes ovarian cancer stem cell activity through ZIP4 up-regulation via nuclear receptor PPARγ [37], suggesting the involvement of LPAR1.” We’re grateful for your comments.
  1. Lane 121 page 4, define ERM.
  • Thank you for your sincere recommendation. We have added the description as follows, “Together with LPAR1, LPAR2 regulates phosphorylation of ezrin/radixin/moesin (ERM) proteins, known as membrane-cytoskeleton linkers, and leads to promote ovarian OVCAR-3 cancer cell migration through cytoskeletal reorganization and formation of membrane protrusions [50].”All revised parts were highlighted in red color in the revised manuscript. We’re grateful for your sincere comments.
  1. If you want to include neuropathic pain and pulmonary fibrosis in this manuscript, change the review title, including other pathologies.
  • Thank you for your professional comments. Since our primary concern is cancer, we have re-written the whole session of neuropathic pain to focus only on the ATX-LPAR axis in chemotherapy-induced neuropathic pain. Regarding pulmonary fibrosis, it appeared only in the part of clinical trials and after the discussion of cancer trials. Since discussing clinical trials is one of our primary topics, we cannot avoid it, particularly most past and ongoing clinical trials focus on it. But, pulmonary fibrosis is not our major topic. Herein, we only mentioned it limited in the small part of clinical trials. We hope to maintain our title and sincerely appreciate your understandings.
